# Laboratory Evaluation of Mechanical Properties of Draupne Shale Relevant for CO$_2$ Seal Integrity

**Magnus Soldal** [1,*] **, Elin Skurtveit** [1,2] **and Jung Chan Choi** [2]

1 Department of Geosciences, University of Oslo (UiO), Sem Sælands Vei 1, 0371 Oslo, Norway; Elin.Skurtveit@ngi.no
2 Norwegian Geotechnical Institute (NGI), Sognsveien 72, 0806 Oslo, Norway; JungChan.Choi@ngi.no
* Correspondence: magnus.soldal@ngi.no; Tel.: +47-4112-9454

**Abstract:** The mechanical integrity of caprocks overlying injection formations is one of the key factors for safe storage of carbon dioxide in geological formations. Undrained effects caused by CO$_2$ injection on strength and elastic parameters should be properly considered in the operational design to avoid fracture creation, fault reactivation and unwanted surface uplift. This study presents results from eleven undrained triaxial compression tests and one oedometer test on the Draupne shale, which is the main caprock of the Smeaheia site in the North Sea, to extract parameters relevant for seal integrity. Tests have been performed on samples oriented perpendicular to and parallel with the horizontal layering of the rock to study the effects of sample orientation relative to the loading direction. Results from undrained triaxial tests showed only minor effects of sample orientation on friction and cohesion. However, when loading during undrained shearing was parallel with layering (horizontal samples), measured Young's modulus was roughly 1.4 times higher than for the vertical samples. Undrained shearing of vertical samples generated 30–50% more excess pore pressure than for horizontal samples with similar consolidation stress owing to more volume compaction of vertical samples. With apparent pre-consolidation stress determined from a high-stress oedometer test, the normalized undrained shear strength was found to correlate well with the overconsolidation ratio following the SHANSEP (Stress History and Normalized Soil Engineering Properties) procedure.

**Keywords:** Draupne; shale; caprock; rock mechanics; seal integrity; SHANSEP

## 1. Introduction

Underground storage of carbon dioxide is considered one of the major mitigation methods to limit global warming. Oil and gas reservoirs and saline aquifers with sandstone or carbonate rock formations overlain by low-permeable shales or evaporites are the main sedimentary basins considered for the geological storage of carbon [1]. The successful injection and storage of CO$_2$ into the subsurface relies on an extensive and robust caprock above the storage formation, ensuring that buoyant and low-viscosity CO$_2$ does not migrate out of the storage complex [2]. Capillary forces prevent CO$_2$ which accumulates below the caprock from flowing through the seal (e.g., [3–5]). As injected CO$_2$ enters pores in reservoir rocks already filled with water or hydrocarbons, low fluid and rock compressibility will cause increased pore pressure reaching the caprock [6]. Although relief can be provided by fluid-extraction wells operating concurrently with the CO$_2$ operation [7], increased fluid pressure is generally expected to arise from the injection.

Increased pore pressure affects the effective stresses in the storage complex, which can lead to irreversible mechanical changes. Shear failure and the possible creation of CO$_2$ migration pathways can occur if shear stresses along given planes in the caprock are sufficiently high. As a conservatory estimate, hydraulic fracturing can initiate if pore pressure increases beyond the least principal in situ stress. Mechanical properties of caprocks above CO$_2$ injection reservoirs are among the parameters determining the magnitudes of both overpressure and vertical displacement following injection and have

been evaluated in several previous studies. Orlic [8] considered both production from and subsequent $CO_2$ injection into a depleted hydrocarbon reservoir since both will induce stress changes to formations inside and outside the reservoir. Constructed stress path diagrams for reservoir and caprock were distinctively different following injection; both normal and shear stress were expected to decrease in the reservoir, whereas for the top seal, the shear stress was expected to increase. Although stress changes in the reservoir were much larger in magnitude compared to the caprock, stresses evolved more critically in the caprock and moved towards the Mohr–Coulomb failure envelope. Accurate determination of caprock strength is therefore important. As in many projects, Orlic [8] was lacking laboratory data for the sealed unit and had to turn to similar lithologies for estimates of caprock strength.

Rutqvist and Tsang [9] used coupled computer codes to study hydromechanical and mechanical changes during injection into a hypothetical sandstone aquifer. In the code, linear elastic and isotropic rock properties were assumed and the potential for rock failure was evaluated based on stress criteria for hydraulic fracturing and fault slip. Considering 30 years of injection with constant injection rate gave a substantial pore pressure increase which was highest in the lower part of the caprock and the upper part of the base rock. Since the total stresses are also increasing in and around the injection aquifer because of injection, the effective stress decrease was only about a third of the increase in pore pressure. Nevertheless, the effective stress changes were enough to cause a 0.6 m uplift of the ground surface 1.5 km above the injection point. By assuming that hydraulic fracturing can occur if fluid pressure exceeded the minimum compressive principal stress, a pressure margin for hydraulic fracturing could be constructed based on in situ stresses and pore pressure changes. In lower parts of the caprock, the pressure margin after 10 years of injection was reduced to almost zero. Furthermore, the pressure margin for unfavourably oriented faults had been exceeded by several MPa.

Hawkes et al. [10] provided a review of geomechanical factors that could affect the hydraulic integrity of caprocks above depleted oil or gas reservoirs utilized for $CO_2$ injection. Concerning the risk of induced shear failure, it was closely related to the development of shear stresses at the interface between reservoir and caprock. As the reservoir expands during injection and the caprock is restricted from lateral deformation, significant shear stresses can result. To quantify the induced shear stress following pore pressure changes, Hawkes et al. [10] considered an analytical solution for an isotropic, elastic half-space overlying a reservoir. The developed shear stress depended on both Young's modulus and Poisson's ratio of the overburden. Generally, the risk was found to increase with large pore pressure changes and low caprock strengths, but it was also shown that increased caprock stiffness gave higher shear stress. Similarly, temperature changes within the reservoir can induce shear stresses that depend not only on the thermal expansion coefficients of the reservoir but also on the elastic parameters of the caprock.

Bao et al. [11] conducted a series of numerical test cases to study the importance of various geological formation properties on pressure generation and ground surface deformation during $CO_2$ injection. They monitored how vertical displacement at the ground surface and pressure at the injection point varied with changes in, among other things, caprock permeability, Young's modulus and Poisson's ratio during continuous injection into a 200-m-thick reservoir. Caprock permeability was important for the injection point pressure development, whereas Young's modulus of the caprock was more significant in terms of the ground surface uplift at the injection well. They concluded, however, that reservoir properties were generally more important than those of the caprock. This contrasts somewhat with reports from Newell et al. [12]. They used inverse modelling together with surface uplift and pore pressure data from In Salah, Algeria, to study the importance of key geomechanical and hydrogeological parameters. Caprock permeability, permeability anisotropy and Young's modulus were found to be very important when trying to match the surface uplift at In Salah. Variations in the layer thickness of reservoir

and caprock between the two studies ([11,12]) were suggested as a possible explanation for the differences by Newell et al. [12].

The studies mentioned above ([8–12]) illustrate that hydraulic and mechanical properties of caprocks during subsurface injection of $CO_2$ into depleted hydrocarbon reservoirs and saline aquifers are important in predicting the overall injection response. Shales are among the most abundant materials in the uppermost layer of the earth's surface and often form the barriers above reservoirs considered for carbon storage [13]. Due to a high content of clay minerals, shales have very small pore sizes and very low permeabilities, making them time-consuming and challenging to study in the laboratory [14]. Mechanical testing of shale needs to combine elements of traditional soil mechanics testing with the stresses usually only encountered in rock testing [15]. Probably related to a historically limited and indirect economic interest in shales, they have not been tested as much as other sedimentary rocks, and there are currently no international standards guiding shale testing [16]. Reliable determination of mechanical properties of shales in the laboratory requires core material prevented from drying and mechanical damage during sampling and storage and test procedures and equipment suited for the purpose (e.g., [13,16–23]).

The Draupne shale is a major caprock of the North Sea and important in the assessment of potential $CO_2$ storage sites in the area [24]. Several recent publications have touched upon specific mechanical aspects of Draupne derived from testing on the same core material as in the current study (e.g., [25–29]). The first mention of the core was by Skurtveit et al. [28] in 2015. Here it was documented that coring and sampling had been conducted in an optimal way to ensure intact samples and that subsequent storage was done appropriately to ensure minimal material drying. Visual inspection after opening three of the nine one-meter long core sections revealed the presence of numerous bedding-parallel unloading fractures and only a few shear fractures [24]. Undrained triaxial compression tests on samples with different sample axis orientations relative to the layering of the rock showed that Draupne is anisotropic in terms of undrained strength. The results from tests on samples oriented perpendicular to and parallel with layering, which was briefly evaluated in Skurtveit et al. [28], are included and discussed in more detail in the current study together with recently performed tests. Koochak et al. [27] made further use of the core material and performed a comparison study between the two Kimmeridge equivalents Draupne and Hekkingen shale from the Barents Sea. They performed uniaxial strain tests (UST) at in situ stress conditions ($\sigma_V' = 26$ MPa and $\sigma_H' = 17$ MPa) with the aim of studying velocity anisotropy and relate potential differences to organic matter content. Despite obvious similarities between the shales (deposited at the same geological time, similar depositional conditions and similar mineralogy), Hekkingen shale has a significantly higher organic content than the Draupne shale. Both shales showed a high and quite similar degree of anisotropy with respect to shear and compressional wave velocities during loading. Anisotropy was seen to decrease with increasing pressure, which was attributed to the decrease in crack density with increasing stress also reported by others (e.g., [30–32]). At high stresses, ultrasonic velocities measured in the Draupne shale were less sensitive to stress changes due to its relatively stiffer framework owing to the lower organic content. Similar anisotropy in terms of compressional wave velocity for Draupne was observed from one undrained triaxial compression test on Draupne material conducted by Grande et al. [26]. After creating a shear fracture on a vertical sample in the triaxial apparatus, they remobilized the fracture at progressively lower confining pressures while monitoring potential acoustic emission. Based on the observed aseismic behaviour of the Draupne shale, they concluded that microseismic monitoring might not be effective in detecting fracturing in the caprock.

Despite the many studies on the Draupne shale mentioned above, a thorough description of its mechanical properties needed for seal integrity evaluation is, to the authors' knowledge, missing in the literature. Therefore, we present results from 11 undrained triaxial compression tests on Draupne samples oriented perpendicular to and parallel with layering. Importantly, the aim is to properly define Draupne failure criteria and provide

experimentally derived values of elastic properties. Finally, one high-stress oedometer test is included to estimate the maximum pre-consolidation stress, which allows for the calculation of normalized undrained shear strength.

## 2. Material

### 2.1. Geological Setting

The North Sea is an intracratonic basin formed within continental interiors and overlying the continental crust. The creation of major sedimentary basins on continental crust is made possible by thinning of the crust and consequent subsidence to maintain isostatic equilibrium. From the Late Carboniferous throughout Late Jurassic times, the North Sea underwent rift periods with stretching and thinning and intermediate periods with thermal cooling and subsidence [33]. Features from these cycles include N-S to NE-SW trending grabens flanked by the East Shetland Basin to the west and the Horda Platform to the east [33,34]. Going from Triassic to Jurassic, the depositional environment in the North Sea shifted from continental to shallow marine. At the same time, the climate became more humid as northwest Europe was forced northward and away from the arid low latitudes. Towards the Late Jurassic, volcanic activity reduced, and the rift systems subsided as the geothermal gradients decreased. Normal faulting along the Viking Graben led to a rotation of fault blocks and exposure of their uplifted shoulders to erosion. Consequently, the Lower–Middle Jurassic strata were removed, and the rift topography was composed of several overdeepened basins [35]. As the sea level rose during the Late Jurassic transgression, poor bottom water circulation caused anoxic conditions when the Draupne Formation (Kimmeridge Clay equivalent [36]) was deposited. The sedimentation rate was relatively high, making the organic-rich Draupne shale both one of the main petroleum source rocks and caprocks in the North Sea [35].

### 2.2. Index Properties

In this study, several meters of Draupne core material have been available for mechanical testing in the rock mechanics laboratory at the Norwegian Geotechnical Institute. The core material was sampled from exploration well 16/8-3S in the Ling depression by former Statoil Petroleum (now Equinor) in 2013 (see Figure 1 for well log data and map of the area). The Ling depression separates the basement highs of Utsira and Sele and acts as a continuation of the Hardangerfjord Shear Zone [33,37]. The well was drilled approximately 40 km east of Sleipner Øst (gas condensate field) and 60 km south of Johan Sverdrup (oil field) in the central part of the North Sea. The Permian reservoir rocks intersected showed poorer reservoir quality than expected, and the well was considered dry. The 9 m of Draupne material sampled from a depth of 2574.5–2583.5 m MD and brought to the laboratory for testing is from the top of the ~85 m-thick Draupne succession in the well.

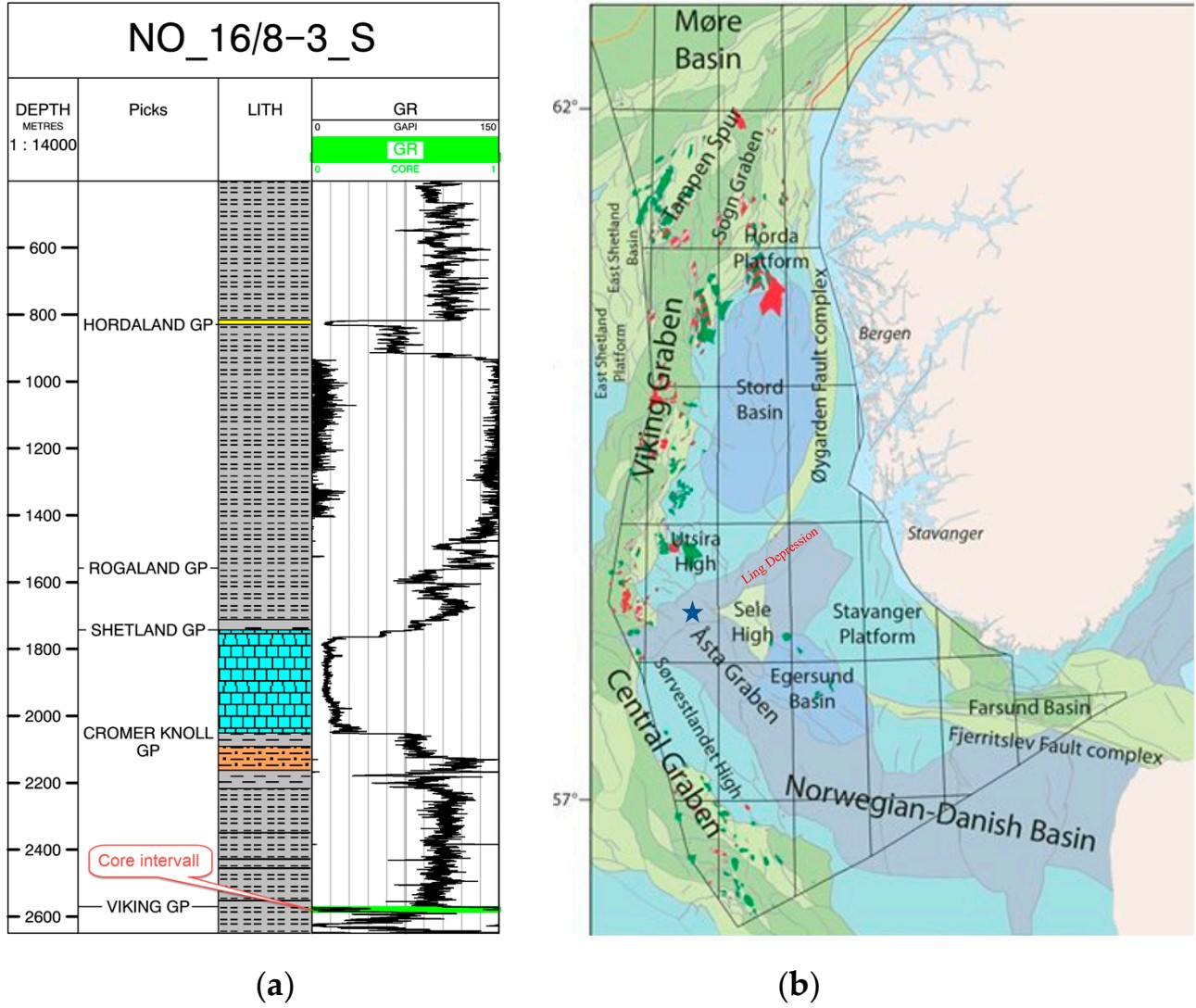

**Figure 1.** (**a**) The gamma-ray log from well 16/8-3S from where the material used in the current study was sampled from Skurtveit et al. [28]. The core interval from the top of the Draupne section is indicated by "core interval" in red. (**b**) Map showing the location of the Ling depression in the North Sea and approximate location of well 16/8-3S (blue star). Map modified from the Norwegian Petroleum Directorate [38].

Characteristic index properties of the Draupne shale have been reported from several previous studies [27–29,39] and are here supplemented with additional bulk density and porosity measurements in Table 1. The initial bulk densities of all samples for triaxial testing were calculated based on initial mass, diameter and height measurements. The samples' initial porosities were calculated using a grain density of 2.49 g/cm$^3$, which is the average between two helium pycnometer grain density measurements performed on cut-offs [28]. The average bulk density was 2.25 g/cm$^3$ (standard deviation 0.03 g/cm$^3$), and the average initial porosity was 15.1% (standard deviation 1%) (Table 1). Porosity estimate from mercury intrusion porosimetry (MIP) gave much lower porosity (6.5%), but as pointed out by Busch et al. [40], MIP tends to overestimate bulk density and underestimate porosity. Determinations of mineralogical composition from X-ray Diffraction (XRD) were performed in three different studies on the core material [27–29] and are here compiled to evaluate the scatter in mineralogy. Table 1 gives the weight percentages of the various minerals found in the Draupne shale. The clay fraction constitutes roughly 50% of the material, and the main clay minerals are kaolinite, smectite and illite. Amounts of organic content (TOC) have both been measured in the laboratory [27–29] and estimated based on resistivity

or density logs covering Draupne occurrences over a larger area [39]. With a consistent average TOC of 6–8 wt.%, Draupne is considered an organic-rich shale [41]. The external specific surface area (excluding the interlayer space of clay minerals) was determined by measuring the adsorbed nitrogen volume of a small, dried and degassed sample [42]. The cationic exchange capacity, expressing the amount of positively charged cations that can be accommodated on the negatively charged surfaces of clay minerals, was determined using the ammonium acetate method [43]. The critical pore diameter can be defined as the smallest pore that completes the first interconnected pathway through the porous network. Data interpretation from mercury porosimetry examination on cut-off samples gave a 9 nm critical pore diameter for Draupne [28] (Table 1). Hydraulic conductivity was measured using the steady-state method on one vertical and one horizontal short sample (2″ diameter) by Skurtveit et al. [28]. The samples were isotropically consolidated to an effective stress corresponding to the estimated in situ effective octahedral stress ($\sigma_{oct}' \approx 23$ MPa) before a constant pore pressure gradient was imposed between the top and bottom of the samples and the flow of water monitored. The coefficient of permeability is, therefore, given as a constant in Table 1, but the stress-dependency is evaluated later from the incremental loading oedometer test performed in the current study. Based on the low permeability, high $CO_2$ capillary breakthrough pressure and rear occurrence of shear fractures, Draupne shale from the core section described herein is expected to be an excellent caprock [28,44].

**Table 1.** The mineralogical composition and selected index properties of the Draupne shale used in the current study ([27–29]).

| Mineralogical Composition (wt.%) | |
|---|---|
| Quartz | 19–25 |
| Feldspars | 7–18 |
| Carbonates | 2–7 |
| Pyrite | 3–13 |
| Total clay | 41–53 |
| Total organic content | 6–8 |
| Bulk density (g/cm$^3$) | 2.25 |
| Initial porosity (%) | 15.1 |
| Surface area (m$^2$/g) | $11 \pm 1$ |
| Cationic exchange cap. (meq/100 g) | 23 |
| Critical pore diameter (nm) | 9 |
| Vertical coeff. of permeability (m/s) | $1.3 \times 10^{-15}$ |
| Horizontal coeff. of permeability (m/s) | $5.8 \times 10^{-15}$ |

## 3. Method

Due to the high content of clay minerals, shales have very small pore sizes and low permeabilities. In addition, water is structurally bound to the minerals, making it challenging to measure the elastic properties of the solid material in shales [14]. Mechanical testing of shale includes elements of traditional soil mechanics testing, but are often performed under stresses usually only encountered in rock testing [15]. Unfortunately, there are currently no international standards targeting shale testing, but several experimental studies in the literature have been conducted in ways that honour the specifics of shales (e.g., [13,16–23]). In the current study, efforts are put into maintaining material saturation throughout the process of core storage, sample preparation and testing and to accurately measure effective stresses during sample deformation.

### 3.1. Sample Preparation

Since even small saturation changes in shales can affect the mechanical behaviour, special care should be taken to preserve in situ saturation. Generally, this involves storing the material submerged in hydrocarbons or in humidity-controlled air and minimizing exposure to air and possible drying [45]. In the current study, we present results both from experiments conducted in 2015 [28] and from new experiments performed in 2020/21. During the approximately 5 years between the two testing campaigns, the material has been kept submerged in oil in a temperature-controlled storage room. Initial water content measured on triaxial samples in 2015 ranged from 6.1–7.3%, with an average value of 6.7%. In 2020–2021, the measured initial water contents were between 6.0–7.3%, again with an average value of 6.7%. Based on the conformity of the initial water contents measured, we therefore assume that the shale material has not suffered from any drying over the years and do not refer to "old" or "new" tests when the results are presented. Except for a 25–30 cm long rubble zone located at the very top of the core section [28], the remainder of the intact core is treated as one (i.e., no reference to relative sample location within the core section is made).

The sample used for oedometer testing in this study was prepared from a core section initially separated from the remaining core using a circular saw. The end surfaces of the core section were then made parallel and planar using a grinding machine. To extract a sample with the exact dimensions of the oedometer cell, a lathe was used in a controlled humidity environment. The height of the oedometer sample was about 20 mm and the diameter 50 mm. Preparation of samples for triaxial testing followed the same procedure up to grinding. After grinding, triaxial samples were sub-cored from the core section using a 1" custom-made drill bit with an internal, air-pressure supplied piston maintaining constant vertical load on the sample during drilling. Both samples with sample axis perpendicular ('vertical samples') and parallel ('horizontal samples') to the horizontal layering of the rock were sub-cored and tested. The triaxial samples had a 25.4 mm diameter and height to diameter ratio ranging from 2 to 2.5. These smaller than normal sample sizes for triaxial testing in rock and soil mechanics were chosen to reduce the testing time, as the allowable strain rate is inversely proportional to the square of the sample size. Despite the reduced sample diameter deviating from that recommended in rock testing standards (e.g., ASTM, ISRM), many examples of shale testing using smaller diameter samples exist in the literature (e.g., [16–18]).

### 3.2. Oedometer Testing

The oedometer apparatus consists of a rigid load frame, a steel oedometric cell and three hydraulic pumps controlling total axial stress and pore pressures in the top and bottom of the sample. Two linear variable differential transformers (LVDTs) measure vertical deformation, whereas horizontal deformation is prevented by the non-compliant oedometric cell (for more information about the oedometer setup used, see [46–48]). Consolidation in the oedometer is consequently one-dimensional, meaning that deformation can only occur in one direction. Since the lateral extent of reservoirs is often much larger than their thickness, one-dimensional consolidation is often considered a reasonable approach [14]. The horizontal stress developed following vertical stress changes is not measured in the oedometer test, and therefore, the mean stress cannot be calculated. Assumptions of the lateral effective stress ratio ($K = \sigma'_h / \sigma'_v$) to calculate the horizontal stress during testing have not been considered here due to the stress path dependency of such ratios [49]. Nevertheless, assuming expulsion of pore water during loading only occurs vertically and consolidation is vertical, the time-dependent one-dimensional settlement in the oedometer can be examined. After the total vertical load is increased as quickly as practically possible in the incremental loading oedometer test, an excess pore pressure develops within the sample. Keeping the total stress constant, the flow from regions of high excess pore pressure to regions of low (and no) excess pore pressure and subsequent pressure dissipation is monitored by changes in vertical settlement over time.

One incremental loading test was performed on a vertical Draupne shale sample. After placing the sample inside the oedometric cell, the porous filters in the top and bottom were flushed with a synthetic brine solution, while the total vertical stress was adjusted to prevent vertical swelling. The synthetic brine was composed of 37 g/L NaCl—the same as the pore fluid salinity in NaCl equivalents determined by Skurtveit et al. [28]. The backpressure was then increased to 0.05 MPa, still under the boundary condition of zero vertical deformation. The recorded effective vertical stress needed to prevent vertical swelling of the sample in contact with brine was about 10.7 MPa. Incremental loading was started from 10.7 MPa effective vertical stress, and a total of 9 load steps were conducted, according to Table 2, over a test period of more than 4 months. The first load step was merely included to adjust the starting stress so that a final, target stress of 76 MPa could be reached in the last loading step. From 15 MPa and onwards, the stress increments during loading were equal to half of the last stress level. This deviation from the normal soil oedometer testing procedure, where each stress level should be double that of the previous, was chosen to enable more load steps before reaching the load capacity of the equipment. Since the piston movement causing the change in vertical stress is hydraulically driven, loading is not fully instantaneously applied. In the load step with the largest change in vertical stress (step 7 in Table 2), loading took almost 2 min to complete.

**Table 2.** Various load steps used in the incremental loading oedometer test on the vertical Draupne shale sample.

| Load Steps during Incremental Loading in Oedometer | | |
|---|---|---|
| **Load Step** | $\sigma_V{}'$ **Start of Load Step (MPa)** | $\sigma_V{}'$ **End of Load Step (MPa)** |
| 1- Loading | 10.7 | 15 |
| 2- Loading | 15 | 22.5 |
| 3- Loading | 22.5 | 33.75 |
| 4- Unloading | 33.75 | 22.5 |
| 5- Reloading | 22.5 | 33.75 |
| 6- Loading | 33.75 | 50.63 |
| 7- Loading | 50.63 | 75.95 |
| 8- Unloading | 75.95 | 50.63 |
| 9- Unloading | 50.63 | 30.75 |

Soft and compressible soils often display a bilinear behaviour when vertical strain is plotted versus the logarithm of effective vertical stress from oedometer testing. The transition is marked by yield stress known as the pre-consolidation stress ($\sigma c'$), and the ratio between pre-consolidation stress and in situ effective vertical stress gives the overconsolidation ratio (OCR) [50]. For sedimentary rocks that have been deeply buried, is more often referred to as the apparent pre-consolidation stress due to ageing and diagenetic processes [21]. In this study, Casagrande's graphical approach was used to locate the apparent pre-consolidation stress [50].

The coefficient of volume compressibility for each load step was calculated as the ratio between vertical strain and change in effective vertical stress. The inverse of one-dimensional volume compressibility is the constrained modulus (M). Permeability and compressibility determine the rate at which water is expelled from the sample, the evolution of excess pore pressure and duration of consolidation. The relationship between hydraulic

permeability ($k_w$), constrained modulus, unit weight of pore fluid ($\gamma_w$) and coefficient of consolidation ($C_v$) is given in Equation (1).

$$C_v = \frac{k_w * M}{\gamma_w} \tag{1}$$

Ferrari et al. [51] showed how poroelastic constitutive equations could and perhaps should be applied to results from incremental loading of shales in oedometer testing to improve the estimation of hydraulic permeability. In the current study, however, only the simpler relationship in Equation (1) has been used to examine the stress-dependency of permeability. The coefficient of consolidation provides a measure of the rate at which consolidation occurs and is related to dissipation of excess pore pressure ($u$) over time ($t$) through a layer with a thickness ($z$) according to the differential equation in Equation (2) [52]:

$$C_v \frac{\partial^2 u}{\partial z^2} = \frac{\partial u}{\partial t} \tag{2}$$

The progress of consolidation can be indicated by the degree of consolidation, which is the ratio of excess pore water dissipated to the initial excess pore pressure generated by the loading. Dissipation of excess pore pressure does not evolve linearly within the sample, and isochrones of equal time show the distribution of degree of consolidation versus distance from the permeable layers [52]. The dimensionless time factor ($T$) relates to elapsed time ($t$), drainage path ($H_{dr}$) and coefficient of consolidation through Equation (3):

$$T = \frac{C_v t}{H_{dr}^2} \tag{3}$$

The time factor corresponding to 50% consolidation is 0.197 [52], and the drainage path for the oedometer test with two-way drainage is half the sample height. Therefore, the coefficient of consolidation can be expressed by the time until 50% consolidation ($t_{50}$) and the sample height ($H$) according to Equation (4):

$$C_v = \frac{0.197 * \left( H/2 \right)^2}{t_{50}} \tag{4}$$

Time to 50% consolidation can be estimated using the logarithm of time fitting method by Casagrande and Fadum [53]. In this method, vertical deformation is plotted against logarithm of time for each load step. One line is drawn through the final points representing secondary consolidation, and another line is drawn tangent to the steepest part of the curve. The intersection of these lines corresponds to deformation and time until 100% consolidation. The deformation at 50% consolidation is then calculated as the midpoint between the corrected zero point and the deformation at 100% consolidation, and $t_{50}$ represents the time of this deformation [54]. Coefficients of consolidation are reported from the consolidation arising from both loading (sample compression) and unloading (sample expansion) in the current study. Differences in $C_v$ measured from loading and unloading could arise from the increased stiffness often measured during loading, and Equation (1) can be used to evaluate the change in hydraulic permeability during the loading sequence.

*3.3. Triaxial Testing*

The triaxial tests were conducted inside a traditional type of pressure cell where a change in cell pressure normally causes an equal change in vertical and horizontal stresses on the sample (schematic of the triaxial cell given in Figure 2). Additional axial load is supplied by a stepping motor located beneath the cell base. The cell is filled with lubricating silicone oil and pressurized using accurate pressure controllers. The top and bottom platens are penetrated by pore pressure tubing, and porous metal filters between the rock sample and platens ensure fluid distribution at the interface. Top and bottom

pore pressure lines are connected to separate pressure sensors and controllers. Horizontal and vertical deformation of the sample is measured inside the pressure cell using LVDTs attached to the membrane but directly in contact with the sample. Two LVDTs used for horizontal deformation measurements are mounted onto light aluminium rings and oriented 90 degrees relative to each other. The LVDTs have spring-loaded cores acting radially towards metallic knobs that penetrate and are glued onto the membrane and record the change in diameter at positions corresponding to $^{1}/_{3}$ and $^{2}/_{3}$ of the sample height. The vertical deformation is measured by another two LVDTs mounted onto the aluminium rings and recording the change in distance between them. Since the diameter of every sample is slightly different, and thus, the membrane stretched to a different degree, the distance between the metal knobs on which the horizontal LVDTs act is measured for every test. This distance, together with the sample height, gives the ratio needed to "scale" up the local vertical deformation measurements so that changes in sample height can be calculated. After failure, the internal and local measurements of deformation are not considered reliable anymore, and further vertical deformation is measured by an externally mounted LVDT. Vertical load is measured by a vented load sensor placed beneath the sample. The influence of cell pressure on the load sensor is small and corrected for. Cell pressure and pore pressure in the top and bottom are measured by 70 MPa electronic pressure sensors. The membrane surrounding the sample is made up of a material that is both relatively soft and at the same time prevents diffusion of water between the sample and the surrounding silicon oil. The "dead volume" between the sample and the valve closed during undrained testing is minimized by using 1/16 " steel tubing; this is important because the pore pressure development can be affected by the volume and compliance of the system.

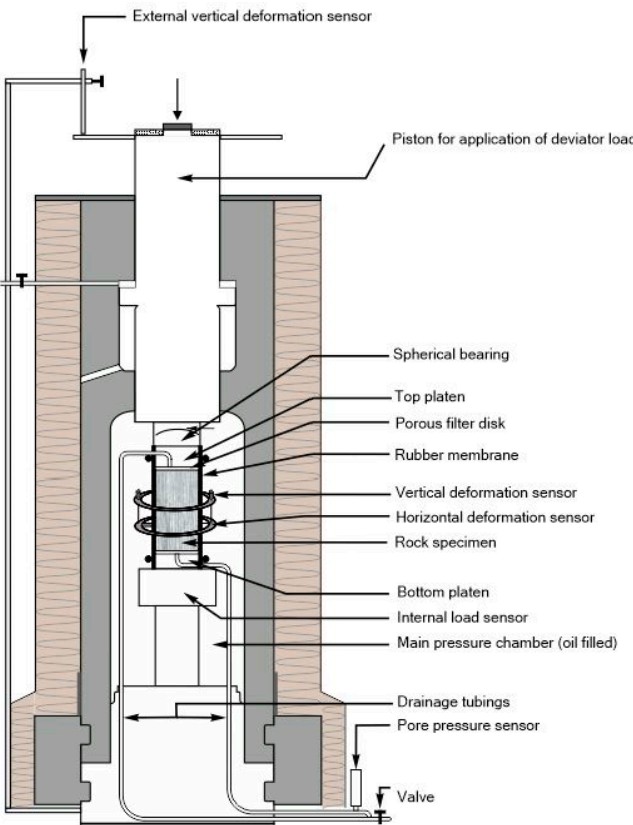

**Figure 2.** Schematics of the interior of the triaxial cell used in the current study.

### 3.4. Strain Rates

One key parameter in mechanical testing of shales is the loading rate. If loading is done too fast relative to the sample permeability and drainage conditions, an excess pore pressure can be generated within the sample that is not measured by the pore pressure sensor located outside the sample. Two different methods for the calculation of appropriate strain rates were evaluated. One applies the theory of consolidation to the problem of dissipation of excess pore pressure during triaxial compression and relates the degree of dissipation, coefficient of consolidation and the time to failure under various drainage conditions [55]. Generally, a 95% degree of dissipation has been shown to acceptably derive strength parameters, and the time to failure can be estimated based on this [56]. The loading rate can then be calculated based on an assumed vertical strain at failure. The other approach is based on equations originally proposed to interpret constant rate of strain oedometer tests [57]. Here, the excess pore pressure generated from shearing a sample with a certain permeability at a certain strain rate can be estimated. If allowed for a maximum excess pore pressure corresponding to 5% of the effective consolidation stress, the strain rate causing such excess pore pressure can be calculated for given drainage conditions. In this approach, strain rate varies with consolidation stress (unless assumptions of similar reduced permeability with increased consolidation stress is made). Since the second approach (described by Berre [57]) gave the lowest and most conservative strain rates, it was chosen in this study to make sure uneven distribution of excess pore pressure was avoided. Vertical strain rates between $10^{-7}$ and $10^{-8}$ s$^{-1}$ was used during undrained shearing of Draupne shale and for drained loading phases load rates down to $5 \times 10^{-10}$ s$^{-1}$ were used. All strain rates were calculated assuming all four side drains connecting the porous filters in top and bottom were working throughout the tests.

### 3.5. Triaxial Test Procedure

Before placing the samples inside the triaxial cell, four vertical side drains were placed between the sample and the surrounding rubber membrane. Next, the pressure cell was closed and filled with silicon oil. During the introduction of synthetic pore water (NaCl = 37 g/L) at the sample ends, two different approaches were used. It is well established that even subtle changes in fluid saturation of shales can potentially cause material swelling or shrinkage, which can alter its structural integrity (e.g., [45,58–61]). After shales have been sampled from the depth and brought to zero total stress conditions at the surface, a negative pore pressure develops within the material. Theoretically, the maximum negative pore pressure that can arise from perfectly elastic unloading of an undisturbed sample is equal to the in situ mean effective stress [62]. Therefore, as samples are given access to free water during re-saturation, measured pressure needed to counteract sample expansion can ideally reflect the in situ mean stress. Vertical and horizontal expansion can simultaneously be prevented by adjusting vertical and horizontal stresses to keep zero volume change. However, previous in-house experience with shales has shown that this approach can give relatively high variation in measured swelling pressure between similar samples. A procedure in which only vertical sample expansion is prevented by adjusting the isotropic pressure has provided more consistent results in the past and was adopted for some tests (tests 7, 8 and 11 in Table 3). Since only vertical deformation was prevented in this approach, slightly different volumetric strains thus developed during re-saturation. For the remaining tests, the confining pressure was increased to the effective consolidation stress before pore water was introduced. These tests experienced different levels of expansion or compression during backpressure and consolidation depending on the consolidation stress relative to swelling pressure.

**Table 3.** The matrix of the triaxial tests and the measurements made. Test ID consists of test type_sample axis orientation with respect to rock layering_effective mean consolidation stress_letter to separate tests with other identical identifiers.

| Test # | | Test ID | $\varepsilon_v = 0$ during Re-Saturation | Drained Isotropic Loading | Uniaxial Strain Loading | Drained Anisotropic Loading | Undrained Shearing |
|---|---|---|---|---|---|---|---|
| 1 | Vertical plugs | CIU_90_20_A | | | | | x |
| 2 | | CIU_90_5_A | | | | | x |
| 3 | | CIU_90_20_B | | | | | x |
| 4 | | CIU_90_30_A | | | | | x |
| 5 | | CIDt_90_30_A | | | | | |
| 6 | | CIUt_90_10_A | | | | | x |
| 7 | | CIU_90_9.3_A | x | x | | x | x |
| 8 | | UST_90_23.9_A | x | | x | | x |
| 9 | Horizontal plugs | CIU_0_5_A | | | | | x |
| 10 | | CIU_0_20_A | | | | | x |
| 11 | | CIU_0_30_A | x | x | | | x |

Next, the backpressure was increased to 30 MPa to ensure complete sample saturation. In this stage, the effective isotropic stress was kept constant while the pore pressure was slowly (approximately 0.5 MPa/hr) increased to 30 MPa. After this, a minimum of two days was given for the sample to stabilize (reach an acceptable rate of change in strain with time). To verify stabilization, a criterion of no more than 10 kPa change in pore pressure during constant confining pressure and closed drainage valves was used [57]. For the "simplest" tests (no 1–6, 9 and 10 in Table 3), undrained shearing was then started at the strain rates given in the previous section. For tests 7, 8 and 11, some deviations in the test procedure were included before starting the undrained shear phase:

- One vertical and one horizontal sample (tests 7 and 11) were isotropically loaded under drained conditions to determine the bulk modulus. Both samples were loaded using a constant stress rate. Using five times higher stress rate for the horizontal sample (0.05 MPa/h compared to 0.01 MPa/h) resulted in approximately equal vertical strain rates in the two tests ($5 \times 10^{-10}$ s$^{-1}$), reflecting the anisotropy in permeability given in Table 1. Secant bulk modulus was calculated as the ratio of change in effective isotropic change over a change in volumetric strain ($\varepsilon_{VOL} = \varepsilon_V + 2 * \varepsilon_H$).
- For test 8, a drained uniaxial strain loading ("k$_0$ loading") phase was included for estimation of the constrained modulus (M) as the ratio between the change in effective vertical stress and resulting vertical strain. With constant pore pressure, the total vertical stress was increased to give a vertical strain rate of $5 \times 10^{-9}$ s$^{-1}$, while the confining pressure was adjusted to prevent horizontal deformation. Uniaxial strain loading was started from an effective isotropic stress of 23.9 MPa and continued until an effective vertical stress of 45.5 MPa was reached.
- Finally, for test 7 the initial phase of shearing was conducted under drained conditions for the assessment of the drained stiffness. Using a low strain rate and open drainage valves, axial loading was continued until 0.1% vertical strain had been reached. At this point, shearing was reversed and unloaded almost back to isotropic stress conditions. Secant-drained Young's modulus was estimated from both loading and unloading as the ratio between effective vertical stress change and change in vertical strain. Similarly, Poisson's ratio was calculated from loading and unloading as the change in horizontal strain over the change in vertical strain.

Undrained Young's modulus and Poisson's ratios were calculated from the initial phase of undrained shearing for all tests (up to 0.2% vertical strain) and between vertical

effective stresses corresponding to 40% and 60% of the recorded strength. Despite the latter method deviating somewhat from relevant standards (e.g., ASTM, ISRM), it was used because it provides a measure that is not to influenced by small strain variations and that covers a part of the stress–strain curve where the rock typically behaves elastically. Young's modulus was determined from plots of deviatoric stress versus vertical strain, assuming cross-anisotropic stiffness and undrained Poisson's ratio of 0.5. The undrained shear modulus ($G_u$) is not sensitive to changes in pore pressure and, as such, is easier to compare between undrained triaxial tests. Tangential values of $G_u$ were calculated at effective vertical stresses corresponding to 50% of the effective vertical stresses at peak according to Equation (5).

$$G_u = \frac{\Delta\sigma_V - \Delta\sigma_H}{2(\Delta\varepsilon_V - \Delta\varepsilon_H)} \tag{5}$$

Shear strength was evaluated according to the Mohr–Coulomb failure theory relating the maximum resistance to shear on a plane of failure ($\tau'_f$) to the apparent cohesion ($c'$), the effective normal stress to that plane ($\sigma'$) and the angle of shear resistance ($\varphi'$) according to Equation (6).

$$\tau'_f = c' + \sigma' \tan\varphi' \tag{6}$$

In the triaxial test, failure is represented by the highest recorded shear stress during testing. Principal stresses at failure ($\sigma_V'$ and $\sigma_H'$) are used to construct Mohr circles of effective stresses, and a failure envelope is drawn on plots of shear stress (($\sigma_V' - \sigma_H'$)/2) versus effective normal stress (($\sigma_V' + \sigma_H'$)/2). The failure envelope intercepts the *Y*-axis (shear stress) at the apparent cohesion, and the slope of the envelope gives the friction angle.

The information provided above is systemized in a test matrix in Table 3, where each test has an ID indicating test type, sample axis orientation relative to the horizontal layering of the rock (90 = vertical sample and 0 = horizontal sample) and effective isotropic consolidation stress.

## 4. Experimental Results

### 4.1. Oedometer

Plots of change in sample height versus the logarithm of time for eight of the nine load steps (Table 2) in the oedometer incremental load test on the vertical sample are given in Figure 3. The change in height is considered positive during compression, i.e., when the sample shortens. The first loading step is not included as it is believed to contain some horizontal expansion related to the filling of the oedometer cell. This could also be the case for the load step between 15 and 22 effective vertical stress, where it is difficult to identify the transition between primary and secondary consolidation. The measured vertical stress needed to prevent the sample from swelling was significantly lower than expected both from in situ stress estimates and measurements from triaxial tests. This is likely due to the inherent difficulties in preparing oedometer (shale) samples that fit perfectly into the steel oedometer cell. Consequently, some horizontal expansion might ocure at low effective vertical stresses. For the remaining load steps, the transition from primary to secondary compression was graphically easier to identify, allowing for the determintion of $t_{50}$ (shown with diamond markers in Figure 3).

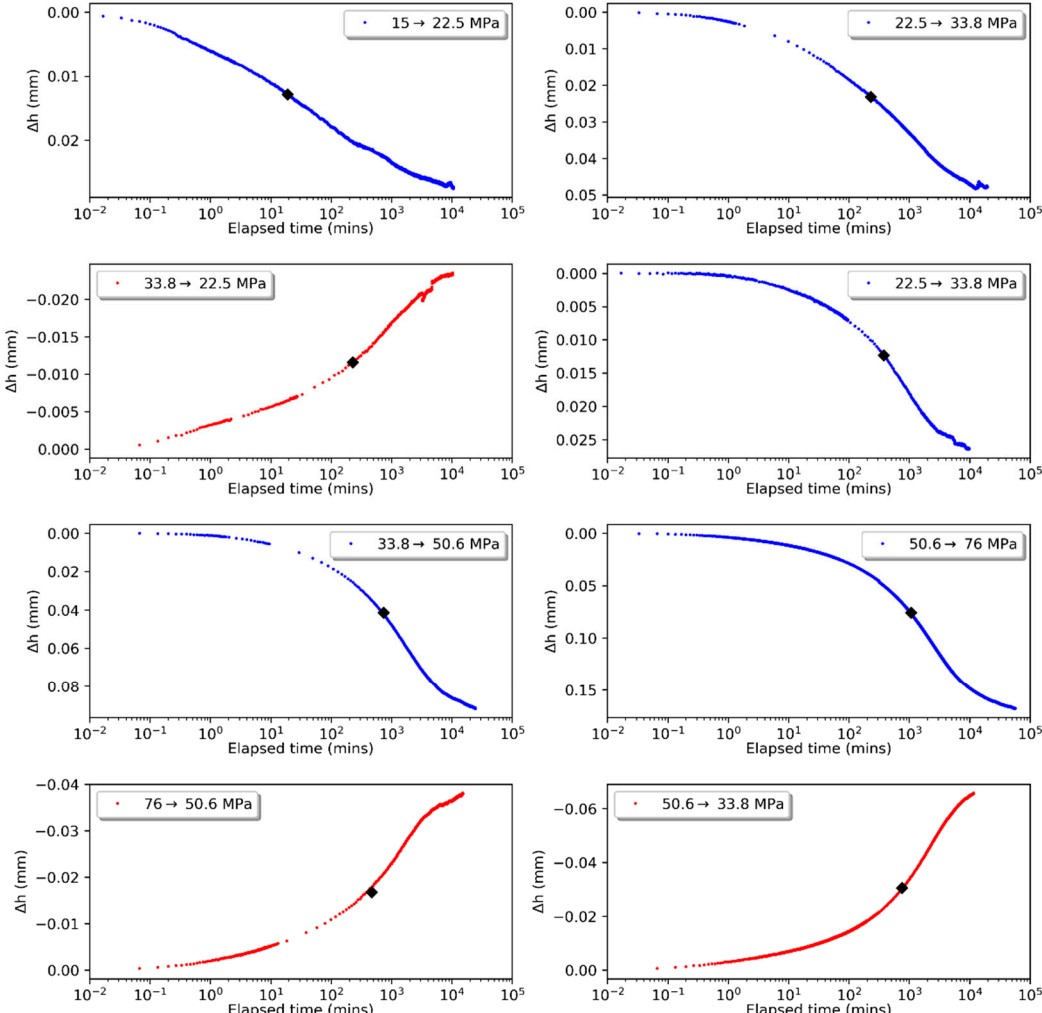

**Figure 3.** The change in sample height (Δh) versus the logarithm of elapsed time from incremental loading on the vertical Draupne shale sample. The magnitude of vertical load change for every step is given in the upper left or right corner of each plot. Blue curves are from loading steps and red curves from unloading steps. Diamond markers indicate $t_{50}$.

The coefficients of consolidation calculated from $t_{50}$ for each load step are plotted against logarithm of effective vertical stress in Figure 4a. Again, the values estimated from loading in the low-stress regime might overestimate $C_v$ since additional radial drainage could be provided between the sample and the radial boundary of the oedometric cell. From $\sigma_V' = 30$ MPa, the coefficients of consolidation from both loading, unloading and reloading are in the range 0.0015–0.0003 mm$^2$/s and decreasing with increasing vertical stress. The constrained modulus (or oedometric modulus) in Figure 4b was calculated from the change in vertical stress over the change in vertical strain for each load step. The loading modulus is between 2.5 and 3.5 GPa for all steps and moderately decreasing with increasing vertical stress. The first unloading modulus after maximum vertical stress is relatively high (8.3 GPa) compared to the next unloading modulus, which is closer to the loading modulus (3.8 GPa). The reloading modulus from 22.5 to 33.8 MPa effective vertical stress is almost twice the loading modulus in the same stress regime.

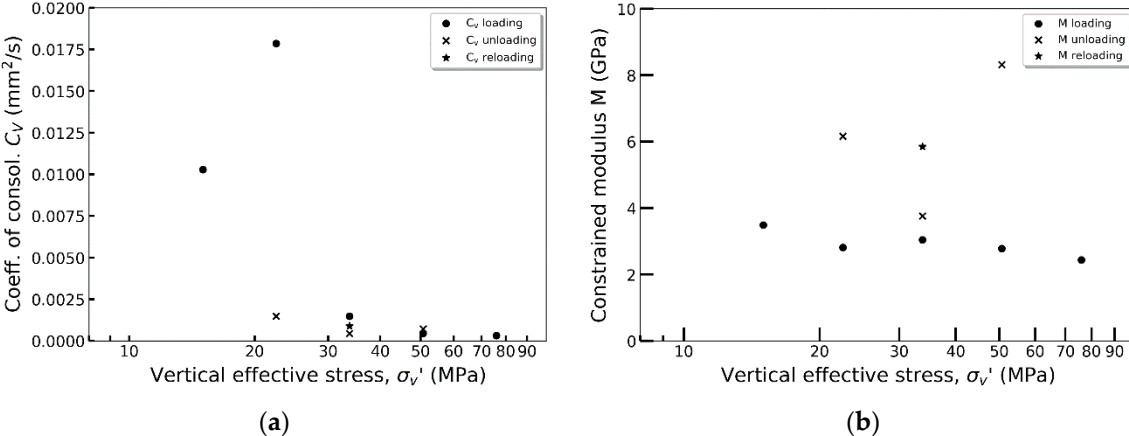

(**a**)                                   (**b**)

**Figure 4.** (**a**) The coefficient of consolidation and (**b**) the constrained modulus versus effective vertical stress from the oedometer incremental loading test of the vertical Draupne sample.

Based on $C_v$, constrained modulus and a water unit weight of 10 kN/m³, the change in permeability was estimated based on Equation (1) and plotted in Figure 5. Except for the first two load steps, permeability coefficients are in the range $0.8–5 \times 10^{-15}$ m/s and decrease with increasing effective vertical stress. As expected, given that the constrained modulus in this test was relatively stress independent, the permeability trend is closely related to the development of $C_v$.

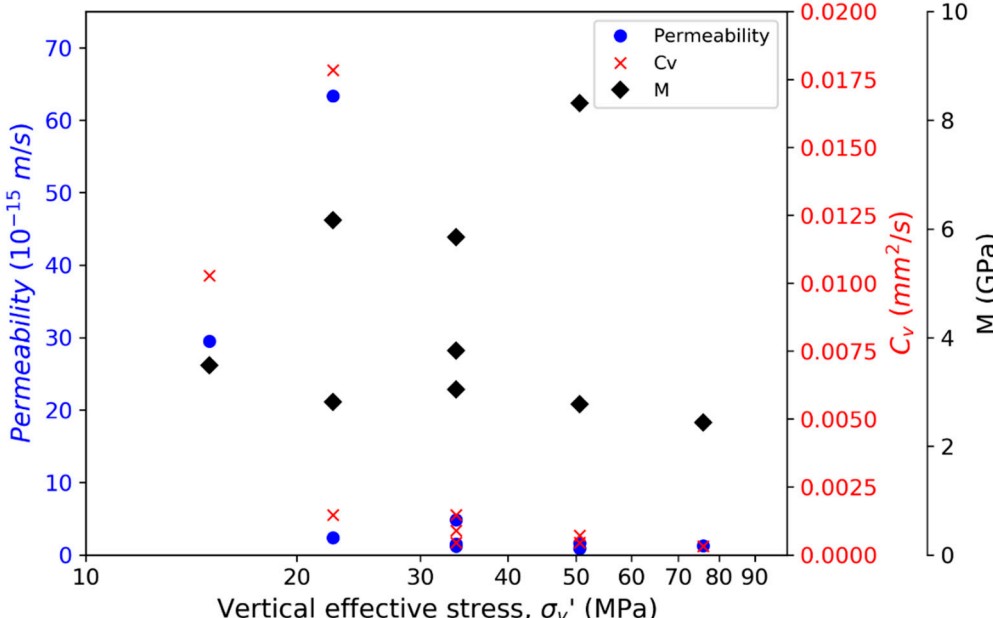

**Figure 5.** The measured coefficient of consolidation (**red**) and constrained modulus (**black**) and calculated hydraulic permeability (**blue**) versus vertical effective stress from incremental loading of Draupne shale in the oedometer.

*4.2. Triaxial Testing-Consolidation*

The measured confining pressures needed to counteract vertical deformation during re-saturation in tests 7, 8 and 11 were 22.3, 23.6 and 23.3 MPa, respectively. In other words, very similar for samples oriented parallel and perpendicular to the rock layering and very similar to the estimated in situ effective octahedral stress of 23 MPa [27]. All samples experienced some volumetric compression during only vertical boundary conditions. For tests where vertical expansion was not counteracted during re-saturation, samples

experienced various magnitudes of expansion depending on the consolidation stress. Potential mechanical effects of this can be qualitatively evaluated by the stiffness and strength results in the following sections.

The recorded volumetric strains during isotropic loading of the vertical (test # 7) and horizontal (test # 11) samples were relatively similar and gave secant bulk moduli of 3.09 and 2.97 GPa, respectively. In the test on the vertical sample, the effective isotropic stress was increased from 24.3 to 27.9 MPa, whereas in the test on the horizontal sample, the effective stress increment was from 30 to 32 MPa. Though the volumetric strains were similar between the two tests, differences in vertical and horizontal strains reflect the samples' orientation relative to the layering of the rock. One sample (test # 8) was loaded under uniaxial strain conditions from a vertical effective stress of about 25 to 45.5 MPa. During this loading, the horizontal effective stress, which was controlled to prevent the sample diameter from changing, increased from 25 MPa to 31.7 MPa. Subsequent unloading back to $\sigma_V' \approx 25$ MPa, caused the horizontal effective stress to decrease to 23.5 MPa. Change in both vertical and horizontal effective stresses are plotted versus vertical strain in Figure 6a. The stress ratios between horizontal and vertical stress during uniaxial strain consolidation ($k_0$) are plotted in Figure 6b both as a stress ratio and ratio of changes in principal stresses. During uniaxial strain loading from isotropic stresses to a vertical stress of 45.5 MPa, the stress ratio went from 1 to about 0.7. The average ratio of stress changes was relatively similar during loading and unloading (around ~0.3–0.4), causing the final stress ratio after unloading to almost reach 1.0 again (ended at 0.95). Technical challenges caused unloading to stop and loading to recommence at approximately $\sigma_V'$ equal 38 MPa, before unloading was again started at a vertical effective stress of 40.3 MPa. The experimental section containing these reversals in the loading direction is removed from the plot of the ratio of stress changes in Figure 6b. Secant-constrained loading and unloading moduli were 3.7 and 4.5 GPa, respectively.

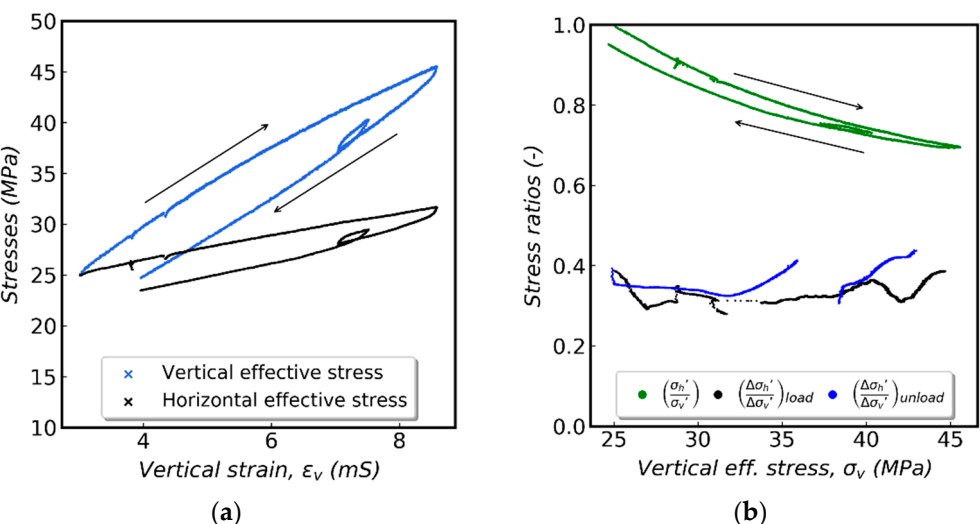

**Figure 6.** (**a**) Stress evolution versus vertical strain and (**b**) the ratio of principal stresses and ratio of change in principal stresses (the black line from loading and the blue line from unloading) versus effective vertical strain from uniaxial strain loading of vertical Draupne sample.

### *4.3. Triaxial Testing-Shearing*

### 4.3.1. Elastic Parameters

Plots of vertical and horizontal strain versus shear stress from three tests on vertical samples and three tests on horizontal samples are given in Figure 7a. Steeper curves, reflecting higher stiffness, were observed for the horizontal samples (0°) compared to the vertical (90°). The vertical strain at failure increases with increasing effective consolidations stress for samples of both principal orientations. The undrained shear moduli calculated

from Equation (5) for all 11 triaxial tests are plotted in Figure 7b. Generally, shear moduli measured for horizontal samples are slightly higher than for vertical samples (average $(G_U)_{hor} = 1.8$ GPa compared to $(G_U)_{ver} = 1.5$ GPa), and in both cases, the moduli are relatively insensitive to effective stress levels.

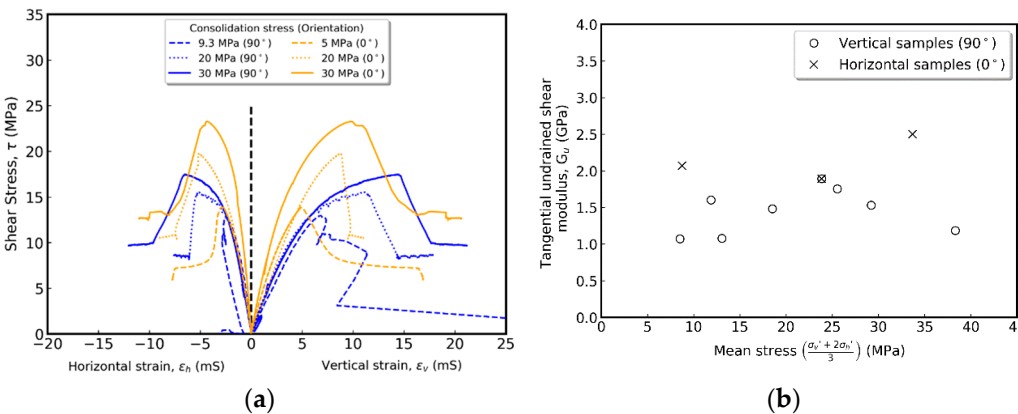

**Figure 7.** (**a**) Vertical and horizontal strains versus shear stress from tests on samples oriented perpendicular to layering (**blue**) and parallel with layering (**orange**). (**b**) Tangential undrained shear moduli.

The undrained Young's moduli were evaluated from plots of deviatoric stress versus vertical strain. Secant values both between 40% and 60% of effective vertical stress at peak ($E_{u,40–60}$) and over the initial 0.2% of vertical strain ($E_{u,ini}$) are plotted in Figure 8a,b versus the average mean stress over which they were measured. Higher values of Young's moduli were recorded for the horizontal samples compared to vertical samples. Secant values between 40% and 60% peak stress for the vertical samples were 4.2–4.7 GPa and between 5.7 and 7.1 GPa for the horizontal samples. No clear stress dependency of Young's modulus was observed for the Draupne samples in this study. Test # 2 (vertical sample tested at low effective consolidation stress) is shown with a red marker in Figure 8a,b since the low Young's modulus ($E_{u,40–60} = 2.4$ GPa) is believed to be influenced by consolidation stress well below swelling pressure and failure to close microfractures during consolidation. Undrained Poisson's ratios were estimated from the ratios of horizontal to vertical strains in a similar way as for Young's modulus (Figure 8c,d). Due to technical problems with one of the horizontal deformation sensors in test # 3, the Poisson's ratio from this test is indicated with a red marker together with test # 2 (low consolidation stress). All tests showed a linear relationship between strains in the principal directions. Poisson's ratios ($\mu_{u,40–60}$) estimated for the vertical samples were from 0.33 to 0.47 (excluding tests # 2 and # 3) and slightly higher for the horizontal samples (0.40–0.54). Note that horizontal deformation is calculated as the average between two deformation sensors oriented 90 degrees relative to each other. For horizontal samples, the sensors are oriented so that they measure parallel and perpendicular to the layering of the rock. If Poisson's ratio is calculated using the individual deformation sensors, the ratio perpendicular to the layering is about three times higher than that parallel with layering. Further analysis of Draupne shale's anisotropic response to loading is beyond the scope of the present work.

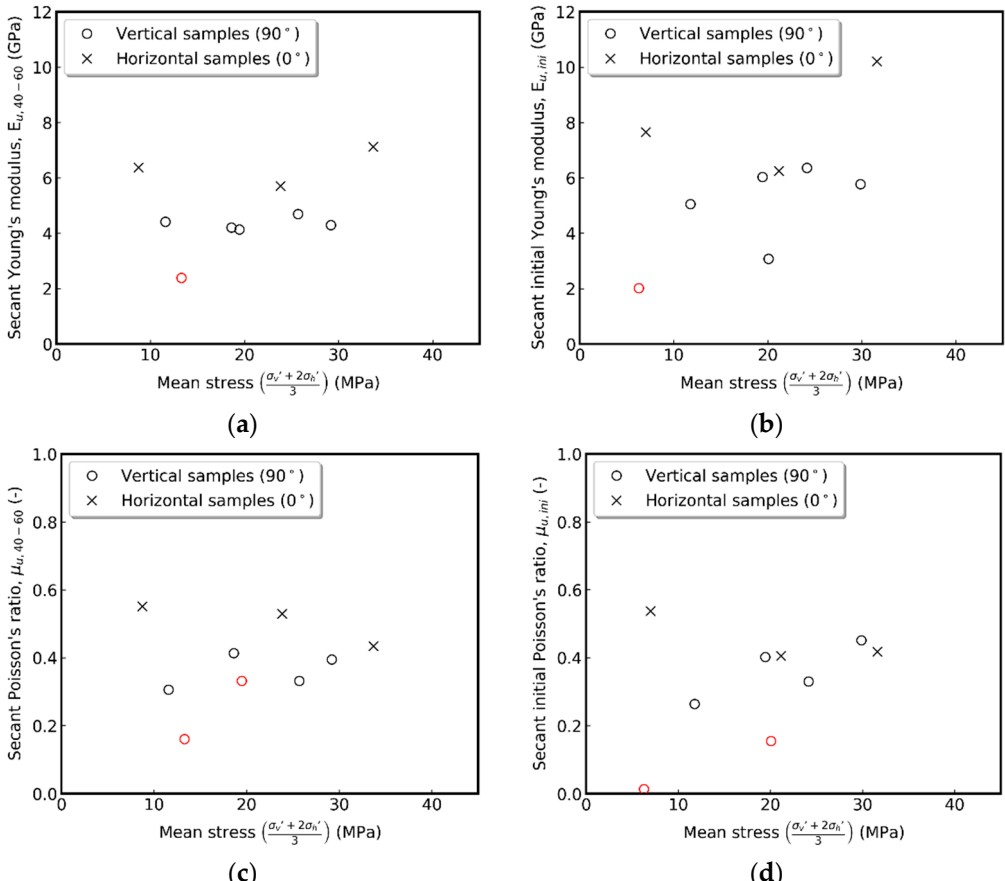

**Figure 8.** (**a**) Secant-undrained Young's moduli between 40% and 60% of vertical stress and (**b**) for the initial 0.2% vertical strain during shearing for vertical (o) and horizontal samples (x). (**c**) Secant undrained Poisson's ratios between 40% and 60% of vertical stress and (**d**) for the initial 0.2% vertical strain during shearing for vertical (o) and horizontal samples (x). Red marker in (**a**–**d**) for the test with low effective consolidation stress likely influenced by swelling during consolidation and for the test with technical issues with a horizontal deformation sensor in (**c**,**d**).

In test # 7, the initial shear phase was done under drained conditions for the measurement of drained Young's modulus and Poisson's ratio. The drained Young's moduli ($E_{d,ini}$) during loading until a vertical strain of 0.1 % and subsequent unloading were 3.7 and 4.6 GPa, respectively. The Poisson's ratio ($\mu_{d,ini}$) measured during drained loading was 0.14 and 0.2 during unloading. When comparing with the undrained stiffness and Poisson's ratio, it should be remembered that the drained measurements were conducted at a lower effective mean stress. As also seen in [16], the unloading step was significantly non-linear, especially shortly after reversing the axial displacement direction. Nevertheless, the values reported herein are secant values for the entire 0.1% vertical strain drained shear phase.

### 4.3.2. Intact and Residual Strength

Shear stresses versus effective horizontal stresses for the vertical and horizontal samples are given in Figure 9. Test # 2, which most likely suffered from poor saturation and mechanical influence of swelling, and test # 5 with drained shearing at elevated temperature are indicated in Figure 9. In all other tests, the pore pressure increased throughout the tests and the highest excess pore pressure was measured close to failure. No indication of temperature effects on the stress path was observed from test # 6 performed under the elevated temperature of 88 °C (shown in orange in Figure 9). The steeper curves for the horizontal samples reflect less volume compression and pore pressure generated, which is also reflected in the higher undrained stiffness measured on horizontal compared to

vertical samples. The onset of dilation, which is often an indication of imminent failure, is virtually absent in all tests.

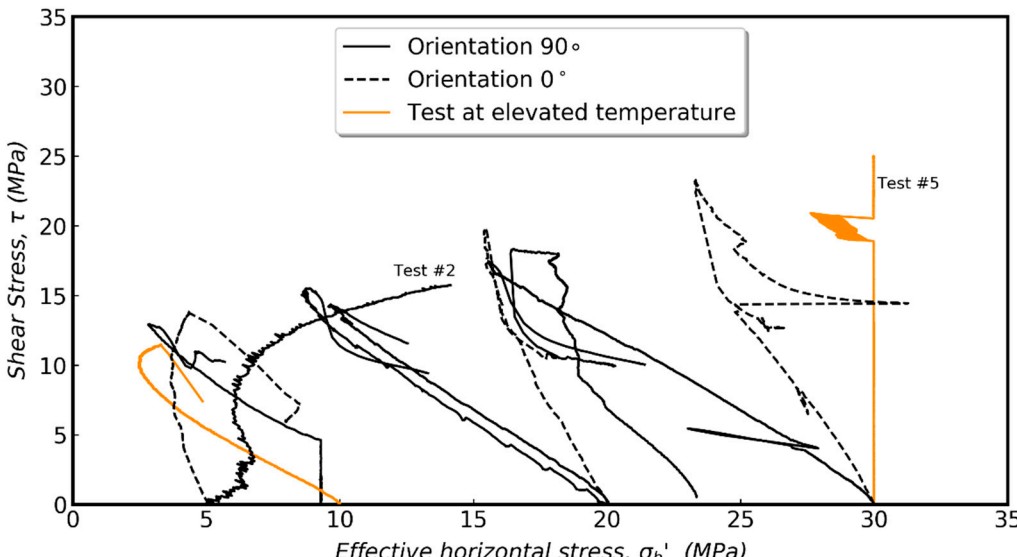

**Figure 9.** Shear stress versus effective horizontal stress during undrained shearing of vertical (solid line) and horizontal (dashed line) samples.

Plots of shear stress versus vertical strain are given in Figure 10. The vertical strain at failure is seen to increase with increasing effective consolidation stress, and shear stresses increase with increasing consolidation stress. Because of the large and abrupt decrease in strength following failure, the actuator initially fails to keep up with the movement along the newly created fracture plane. Therefore, some time is spent after failure before the actuator "catches up", and the residual strength can be measured through continued shearing on the fracture plane. The curves in Figure 10 show that shear stresses stabilized sometime after failure for nearly all tests. This represents the residual strength of the Draupne shale samples tested. For the horizontal sample consolidated to the highest consolidation stress ($\sigma_c' = 30$ MPa), additional phases with reactivation of the fracture plane at lower effective consolidation stresses were added after the first failure.

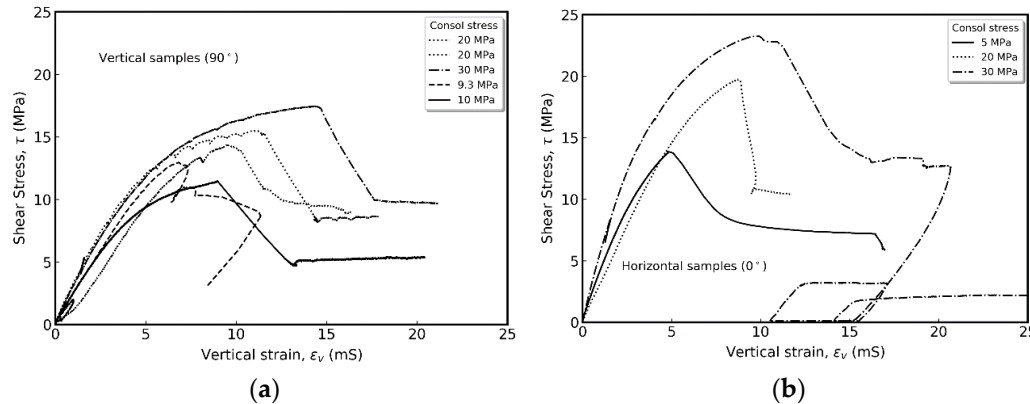

(**a**)                  (**b**)

**Figure 10.** Shear stress versus vertical strain during undrained shearing for vertical (**a**) and horizontal (**b**) samples.

The failure envelopes in Figure 11 are constructed in a manner that iteratively searches for a line minimizing the difference between Mohr circle radii and the distance between the centre of the circle and the failure line. The apparent cohesion and angle of shear resistance (friction angle) of the intact Draupne shale are relatively similar for the vertical

and horizontal samples. The apparent cohesion ranges from 7.5 MPa for the vertical samples to 8.4 MPa for the horizontal samples. Friction is also slightly higher for the horizontal samples—19.4° compared to 18.7° for the vertical samples. Residual strength was possible to extract from seven of the eight tests on vertical samples, and five residual strengths are reported from three tests on horizontal samples (fracture reactivated two times at lower effective consolidation stress for one test). Inspection of samples after testing showed that well-defined fracture planes had developed. Despite some deviation from linearity observed at low stresses when constructing Mohr circles for the residual strength parameters, a linear Mohr–Coulomb (MC) criterion was also used to derive residual strength parameters. The linear MC criterion is considered a well-established criterion suited for the extraction of intact and residual cohesion and friction and for comparing strength anisotropy. Furthermore, the drift from linearity observed at low stresses for the residual, horizontal strength in Figure 11d could also be influenced by the fact that these strengths are measured during the re-activation of the failure plane. Residual friction angle is slightly lower than intact friction angle, while the reduction in cohesion from intact to residual is relatively large.

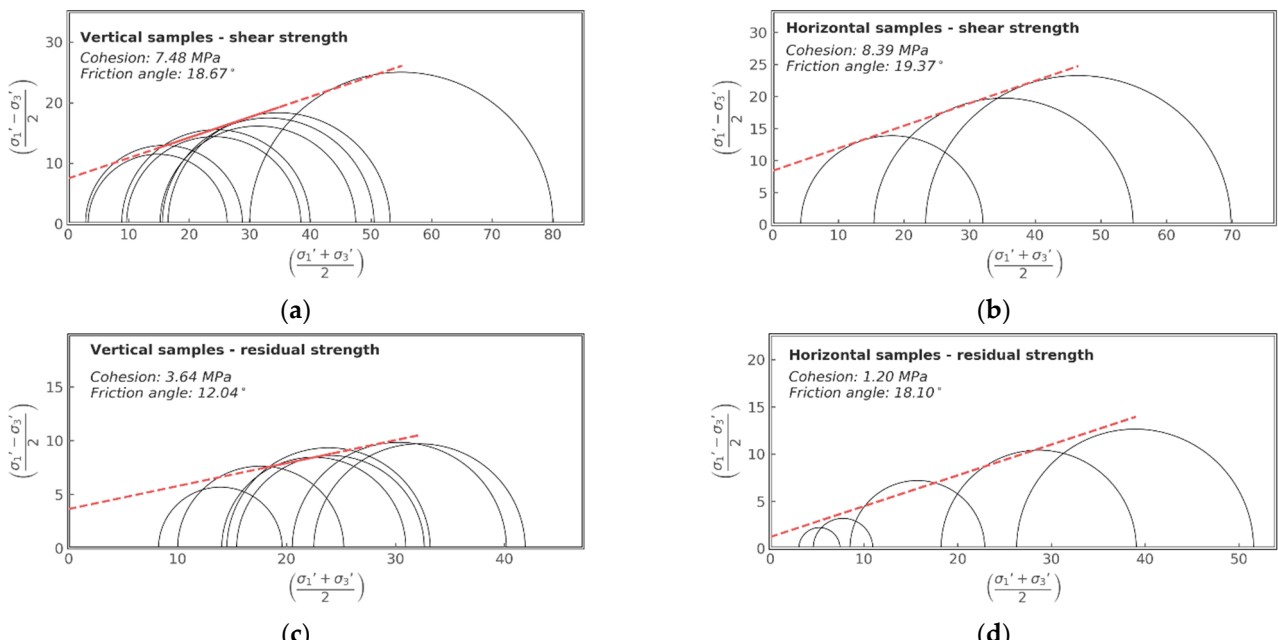

**Figure 11.** The intact (**a**,**b**) and residual (**c**,**d**) apparent cohesion and friction angle from undrained triaxial testing on vertical (**left**) and horizontal (**right**) samples of Draupne shale.

The range of values measured from the current test campaign presented here is summarized in Table 4. It is emphasized that the values reported are measured under a certain stress regime presented herein, and stress sensitivity should be considered when values are to be used.

**Table 4.** The range of values measured from oedometer and triaxial testing of Draupne shale in this study.

| | Parameter | Vertical Samples | Horizontal Samples |
|---|---|---|---|
| **Oedometer** | Coefficient of consolidation (mm$^2$/s) | 0.0003–0.0015 | |
| | Constrained modulus, M (GPa) | Loading: 2.5–3.5 | |
| | | Unloading: 3.8–8.3 | |
| | Apparent pre-consolidation stress (MPa) | 36.5 | |

**Table 4.** *Cont.*

| | Parameter | Vertical Samples | Horizontal Samples |
|---|---|---|---|
| Triaxial | Bulk modulus, $K_b$ (GPa) | 3.09 | 2.97 |
| | Undrained Young's modulus, $E_u$ (GPa) | 4.2–4.7 | 5.7–7.1 |
| | Undrained Poisson's ratio, $\mu_u$ | 0.33–0.47 | 0.40–0.54 |
| | Drained Young's modulus, E (GPa) | Loading: 3.7 | |
| | Drained Young's modulus, E (GPa) | Unloading: 4.6 | |
| | Drained Poisson's ratio, $\mu$ | Loading: 0.14 | |
| | Drained Poisson's ratio, $\mu$ | Unloading: 0.20 | |
| | Undrained shear modulus, $G_u$ (GPa) | 1.1–1.9 | 1.4–2.0 |
| | Apparent cohesion (MPa) | 7.48 | 8.39 |
| | Friction angle (°) | 18.7 | 19.4 |
| | Residual apparent cohesion (MPa) | 3.6 | 1.2 |
| | Residual friction angle (°) | 12.0 | 18.1 |

## 5. Discussion

Shales are usually considered vertical transverse isotropic, meaning that properties are identical in the horizontal plane but different in the vertical direction perpendicular to horizontal layering (e.g., [14,63–65]). The results presented herein allows for a qualitative assessment of the directional importance of some of the Draupne shale properties investigated. Both friction and cohesion were seen to vary only slightly between horizontal and vertical samples. For neither undrained shear modulus nor Poisson's ratio, the results show clear orientational or stress dependency. However, both stress level and orientation are strongly connected to the development of excess pore pressure during undrained shearing. For vertical and horizontal samples consolidated to the same effective isotropic stress, the maximum excess pore pressure was 1.5 higher for the vertical samples and increasing with consolidation stress. Excess pore pressure generation was closely related to less volumetric strain and a significantly higher undrained Young's modulus for horizontal samples. The undrained Poisson's ratio was almost 3-times the drained, and the undrained Young's modulus was about 1.6-times the drained modulus for vertical samples in the current study.

The estimated permeabilities from incremental loading in the oedometer (Figure 5) show good agreement with the steady-state hydraulic permeability measurements made under isotropic stress conditions (Table 1). After the initial loading and unloading up to $\sigma_v' = 22.5$ MPa, further loading to an effective vertical stress of 75 MPa only reduced permeability by approximately 15%. The one-dimensional compressibility measured in the oedometer showed relatively little stress sensitivity, suggesting that reduction in hydraulic permeability was more related to the reduction in coefficient of consolidation. Secant constrained moduli from oedometer testing are compared with both secant and tangential constrained moduli from the uniaxial strain test (test # 8) in Figure 12. Initial tangential modulus from triaxial is relatively high, which may be related to the "unrealistic" stress state at the beginning of uniaxial strain loading (i.e., isotropic conditions and lateral effective stress ratio K = 1). As vertical effective stress increases, modulus decreases and approaches secant values from the oedometer test. Secant modulus from the triaxial is slightly higher than from the oedometer due to the reasons just described. Because of time constraints, uniaxial strain loading was stopped before a constant value of stress ratio between the horizontal and vertical effective stresses had been reached. The final recorded stress ratio before unloading was 0.7 and decreasing, which is higher than the in situ stress ratio of 0.66 based on in situ stresses of $\sigma_V' = 26$ MPa and $\sigma_H' = 17.2$ MPa [27]. Both stress ratios of 0.66 and 0.7, however, are within the expected range for normally consolidated clays [55]. Based on the plots of vertical strain versus effective vertical stress during

uniaxial strain consolidation on a vertical Draupne shale sample by Koochak et al. [27], it is possible to calculate a secant constrained modulus. Loading from the in situ $\sigma_V'$ of 26 MPa to $\sigma_V' = 49$ MPa resulted in about 0.7% vertical strain. The resulting secant modulus is 3.3 GPa which is close to the reported values in the current study (3.7 GPa from uniaxial strain testing in triaxial and 2.5–3.5 GPa from oedometer incremental loading). Considering the uniaxial strain loading from in situ $\sigma_V'$ up to about 45 MPa from Koochak et al. [27] (loading between 45 MPa and 49 MPa effective vertical stress showed a peculiar development in stress ratio and is therefore not included here), the effective horizontal stress changed from 17 to 22.5 MPa. The corresponding ratio of change in effective horizontal stress to change in effective vertical stress is 0.29. This is very similar to the ratio measured herein, even though uniaxial strain loading was started from K = 0.66 in their study and from K = 1 in the current work.

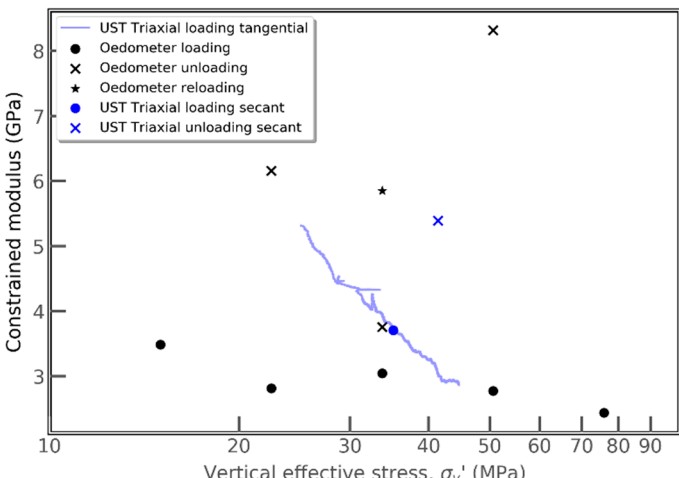

**Figure 12.** A comparison between secant constrained modulus from incremental loading in the oedometer (black) and secant (blue markers) and tangential (blue line) constrained modulus in the uniaxial strain triaxial test.

The construction of Mohr–Coulomb failure criterion using effective stress circles (Figure 11) gave a well-defined failure envelope for both the intact vertical and horizontal samples. There is a tendency for deviation from linearity at low effective stresses. At low stresses, below the stresses needed to counteract sample expansion during re-saturation, deviations could possibly be linked to loss of mechanical integrity following swelling. Very low undrained stiffness measured on the vertical sample with the lowest effective consolidation stress suggests the same. The failure envelope for the residual strength was determined from effective stress circles extending further into the low-stress regime. Consequently, the deviation from linearity is more pronounced here.

Figure 13 shows vertical strain plotted against the logarithm of vertical effective stress for the complete oedometer incremental load test. Markers in the plot indicate the last recording before the next loading step was initiated. Casagrande's graphical approach was used and gave an apparent pre-consolidation stress of 36.5 MPa. Despite the subjectiveness involved in the approach, the apparent pre-consolidation stress interpreted from the oedometer test is within the expected range when also accepting the possible interpretation bias arising from diagenetic processes.

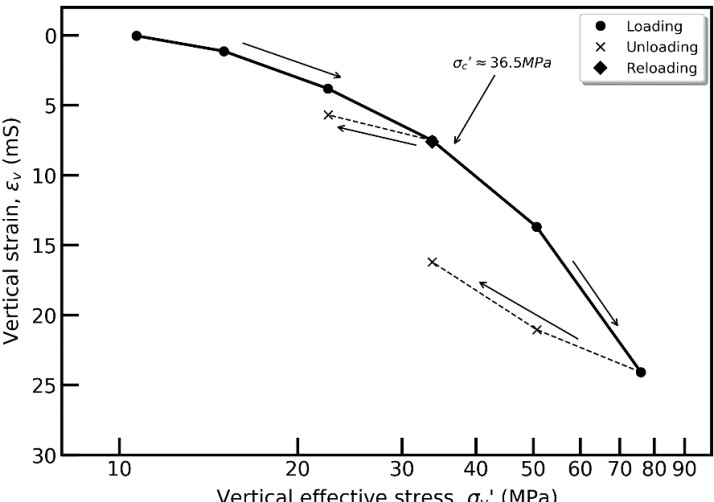

**Figure 13.** The vertical strain plotted against the logarithm of vertical effective stress for the oedometer incremental loading test on a vertical Draupne sample. Markers indicate the last point of a load step before the next loading, reloading or unloading step.

Inspired by the SHANSEP (Stress History and Normalized Soil Engineering Properties) procedure for normalizing the undrained shear strength of clays [66], a similar approach has been explored for clay shales by Gutierrez et al. [13]. According to this approach, the undrained shear strength ($S_u$) of normally consolidated clays is unique when normalized with respect to the current effective vertical stress ($\sigma'_{VO}$). For overconsolidated clays (OCR >1 = maximum past effective stress exceeding current effective vertical stress), the normalized undrained shear strength can be given by Equation (7):

$$\frac{S_u}{\sigma'_{VO}} = a(\text{OCR})^b \tag{7}$$

where $a$ is the normalized undrained shear strength when OCR = 1, and $b$ is an empirical exponent. Gutierrez et al. [13] evaluated OCR based on experimentally determined apparent pre-consolidation stress, acknowledging the fact that diagenetic processes in clay shales affect overconsolidation and strength. They performed undrained triaxial testing on Kimmeridge shale and a Barents Sea shale and calculated OCR from apparent pre-consolidation stress and vertical effective consolidation stress for each test. The highest shear stress recorded during testing was then normalized with respect to the effective vertical stress and $a$ and $b$ parameters found from power regression. By using the apparent pre-consolidation stress from oedometer incremental loading ($\sigma_c' = 36.5$ MPa), we estimated OCR for the triaxial tests on vertical samples in the same manner. Regression gave $a$ of 0.51 and $b$ of 0.72, which is compared to values for Draupne and Kimmeridge provided by Gutierrez et al. [13] in Table 5. They also showed an approximately linear correlation between the logarithm of normalized undrained shear strength and logarithm of OCR for 25 types of shales, indicating that SHANSEP could provide rough estimates of undrained shear strength in the absence of laboratory testing. Due to strength anisotropy, the normalized undrained shear strength correlation with OCR is only valid when loading is with the major principal effective stress perpendicular to layering.

Both Horsrud et al. [67] and Økland and Cook [68] dealt with mechanical characterization of North Sea shales related to borehole stability problems during offshore drilling. Due to scarcity of available Draupne material for testing, Økland and Cook [68] turned to an analogous organic-rich Jurassic outcrop shale from northern England for most of their testing. Data from unconfined compressive strength tests were fitted to a single-plane-of-weakness model which gave a bulk friction angle of 20° and bulk cohesion of 6.0 MPa. Horsrud et al. [67] performed undrained triaxial tests on "unnamed" Tertiary to Triassic

shales from the North Sea. From the Jurassic shales tested they found apparent cohesion values ranging from 4 to 13.5 MPa and friction angles from 7 to 27°. Reported undrained Poisson's ratios were between 0.13 and 0.24 and undrained Young's modulus from 1.4 to 3.8 GPa.

**Table 5.** Power regression fit parameters when normalized undrained shear strength is plotted versus OCR for Draupne tests in this study and for Draupne, Kimmeridge and a compilation of 24 different types of shale in Gutierrez et al. [13]. The referenced Draupne and Kimmeridge were not given with sample depths, but the max burial depth and apparent pre-consolidation stress of Kimmeridge were 1.7 km and 22 MPa, respectively.

| Material | *a* | *b* | $R^2$ | Source |
|---|---|---|---|---|
| Draupne, Ling depression | 0.51 | 0.72 | 0.95 | Current study |
| "Draupne, North Sea" | 0.49 | 0.65 | 0.99 | [13] |
| "Kimmeridge, Dorset, UK" | 0.47 | 0.66 | 0.99 | [13] |
| "25 different types of shales" | 0.37 | 0.87 | 0.8 | [13] |

## 6. Conclusions

Results are presented from one oedometer and eleven triaxial compression tests on vertical and horizontal samples of the Draupne shale aimed at deriving mechanical caprock properties relevant for seal integrity evaluation during CCS operations in the North Sea. Important findings from the laboratory study are:

- For the intact vertical samples, an apparent cohesion of 7.5 MPa and a friction angle of 18.7° were estimated based on constructed Mohr Coulomb failure envelope. Relatively similar values were found for the horizontal samples (cohesion 8.4 MPa and friction 19.4°). In terms of residual strength, the constructed failure envelopes showed a significant reduction in cohesion, whereas the reduction in friction angles was less significant. Both intact and residual strengths are important in the operational design of $CO_2$ injection operations to avoid fracture creation or reactivation of pre-existing weakness planes.
- Secant-undrained Young's modulus measured between 40 and 60% of peak stress was between 4.2 and 4.7 GPa for vertical samples and 5.7–7.1 GPa for horizontal samples. The average secant Poisson's ratios for samples of both principal directions were close to 0.5.
- Whereas the measured constrained modulus from oedometer testing showed little or no stress dependency, the coefficient of consolidation decreased with increasing effective vertical stress. Hydraulic permeability estimated from constrained modulus and coefficient of consolidation therefore also decreased with increasing effective vertical stress. Increasing effective vertical stress from 33 to 75 MPa caused a reduction in hydraulic permeability of around 15%.
- The apparent pre-consolidation stress determined from oedometer incremental loading was 36.5 MPa. Consequently, the tested Draupne shale is considered normally consolidated to lightly overconsolidated. The measured undrained strength from triaxial testing normalized against the effective vertical consolidation stress correlated quite well with overconsolidation ratio following the SHANSEP normalization procedure developed for clays.

**Author Contributions:** Conceptualization, M.S. and E.S.; Data curation, M.S.; Formal analysis, M.S.; Funding acquisition, E.S.; Investigation, M.S.; Methodology, M.S. and J.C.C.; Project administration, E.S.; Validation, J.C.C.; Writing—original draft, M.S.; Writing—review and editing, E.S. and J.C.C. All authors have read and agreed to the published version of the manuscript.

**Funding:** The testing and analysis have been performed with support from the Norwegian CCS Research Centre (NCCS), performed under the Norwegian research program Centres for Environmental-friendly Energy Research (FME), grant 257579/E20.

**Data Availability Statement:** Experimental data supporting the analysis presented herein are published at DataverseNO. Soldal, Magnus, 2021, "Replication Data for: "Laboratory Evaluation of Mechanical Properties of Draupne Shale Relevant for CO2 Seal Integrity"". Data is contained within the article.

**Acknowledgments:** The authors would like to thank the partners in PL360 with funding from CLIMIT (Project 223122) and Statoil (Equinor) for the core material used in this study. Thanks also to Bjørnar Slensvik for help with sample preparation and to Toralv Berre for instructive discussions during the preparation of this manuscript.

**Conflicts of Interest:** The authors declare no conflict of interest.

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
