# Peer review of "Laboratory Evaluation of Mechanical Properties of Draupne Shale Relevant for CO2 Seal Integrity"

_geosciences, doi:10.3390/geosciences11060244_

Round 1

Reviewer 1 Report

Page 5: Quality of Figure 1 can be improved. The Gamma ray log in Figure 1 is difficult to read. The author should consider labelling the Figure 1(a) for the Gamma Log and Figure 1(b) for the map, instead of indicating Left and Right.

Page 7. Lines 281 to 290: (1) There are larger Oedometer Cell available. Is there any reason to use the cell size of “20 mm in height and 50 mm in diameter”? (2) Most triaxial tests are performed on undisturbed soil samples. The diameter of triaxial soil samples normally range from 38 mm to 100 mm. Here, the author had presented a very small triaxial sample with a diameter of 25.4 mm. Why is this sample size selected?

Pages 7 and 8. Lines 311 to 325 and Table 2. The test commenced with an applied load of 10.7 MPa. It is not clear to the reader how the load increment is determined. Why were Step 1 of the test concluded at 15 MPa, and why were Step 2 of the test concluded at 22.5 MPa?

Page 8. Lines 349 to 372. Equations 1 to 4 are fundamental equations from Terzaghi’s Consolidation theory. Not sure the reason for the authors to include these in the manuscript. Particularly, Equations 2 to 4, which are fundamental knowledge (in Soil Mechanics for Geotechnical Engineers) and can be replaced with a simple paragraph with citing the book by Terzaghi (reference [52] in the manuscript).

Page 11. Lines 479 to 486. What is the difference between the vertical and the horizontal samples? How is the horizontal sample being prepared? Is it related to the layering of the rock (line 524)? The author should explain the vertical and the horizontal samples.

Page 11. Lines 503 to 510. Equation 5. In the text, the authors are referring to the calculation of effective vertical stress. However, Equation 5 (line 511) is the same as Equation 4 (line 372). The author should check the manuscript again to make sure Equation 5 is correctly presented.

Page 12. Table 3. B-value is normally measured during a triaxial test to determine the degree of saturation. It is very strange that only certain test that the B-value is measured.

Page 13. Figure 2. During the unloading stage of oedometer test, the rock sample has expanded (or heaved) and this vertical deformation (e.g. 33.8->22.5 MPa) is not settlement. Also, the author should consider changing the line colour for the settlement plot of “22.5->33.8 MPa”.

Page 13. Lines 552 to 563. Page 14. Figure 3. “c /subscript v” is the “coefficient of consolidation”. The authors should be consistent in using this term, and not using “consolidation coefficient”.

Page 14. Figure 3 (a). Coefficient of Consolidation is estimated from time-settlement curve. Not during the unloading.

Page 15. Figure 5. There are many solid and dashed lines in Figure 5. The results seem to be form different tests. The authors should consider using different line or symbol to represent different test. Also, one of the dashed lines seems to have pore pressure dropped during the test. Is it an error?

Page 16. Figure 6. The stress-strain plot in Figure 6 (a) seems very strange. There are points plotted in the Figure 6 (a) (dots moving vertically upward) at vertical strain of 4. Also, during the unloading process, is the unloading stress (for horizontal stress) less than the originally applied stress of 25 MPa?

Figure 22. Figure 13 seems very strange. The authors should check the figure again. By looking at the Blue Line, it appears to have a sudden change in the vertical strain (for example, between vertical stress of 30 MPa to 50 MPa, the blue line shows a slow change in the strain and an abrupt change in the vertical strain when the pressure is at 50 MPa).

Page 25. Line 1002. Missing information for Reference [46]. Which journal is this paper being published?

Reviewer 2 Report

The paper presents an experimental study of the Draupne shale which is a potential caprock for a CO2 reservoir. One consolidation and several triaxial tests were completed and results were well described and discussed. I found that the paper was well written with detailed description of sample preparation, testing procedure and result discussions. I have several questions for improvement:

1) Line 273: Provide evidence to show that the water content and effective pressure are selected to counteract the swelling in this study compared with that in 2015.

2) Line 381: an image of the test setup and instrumentation will be helpful.

3) Figure 9: Discuss the effect of elevated temperature on the pore pressure build-up.

4) Why a nonlinear failure criterion was not applied to define the shear strength?

5) Why secant Young's modulus was determined at 40 to 60% of the peak stress? why the Young's modulus was not taken at the linear portion of the axial stress-strain curve suggested in the ASTM standard?

Round 2

Reviewer 1 Report

The authors have undertaken significant work to present their results in the revised manuscript. The authors have good knowledge on their work.